# Effect of Layer Orientation and Pore Morphology on Water Transport in Multilayered Porous Graphene

**DOI:** 10.3390/mi13101786

**Published:** 2022-10-20

**Authors:** Chulwoo Park, Ferlin Robinson, Daejoong Kim

**Affiliations:** Department of Mechanical Engineering, Sogang University, Seoul 04107, Korea

**Keywords:** carbon, graphene, porous nanosheet, multilayered graphene, water transport, layer orientation, pore morphology

## Abstract

In the present work, the effects on water transport due to the orientation of the layer in the multilayered porous graphene and the different patterns formed when the layer is oriented to some degrees are studied for both circular and non-circular pore configurations. Interestingly, the five-layered graphene membrane with a layer separation of 3.5 Å used in this study shows that the water transport through multilayered porous graphene can be augmented by introducing an angle to certain layers of the multilayered membrane system.

## 1. Introduction

The demand for freshwater is increasing day by day due to the increase in population and pollution of existing freshwater sources. Desalination, the process that removes the salt and other heavy metals from saline water, has become the prominent technology to address the above-mentioned issue. Researchers have perceived that the size of a substance can influence its physiochemical properties [1,2]. This paved the way for increased subsequent research into nanomaterials. Due to the advancement of technology, the manufacturing and application of nanomaterials in the desalination process have become more common [3]. On the other hand, the nanomaterials show excellent physical and chemical properties [4], which makes them ideal candidates for the desalination process. Graphene is one such potential candidate for the desalination process [5]. Conventionally, graphene is considered a hydrophobic material [6]. Some recent studies claim that graphene is hydrophilic and exhibits a low water contact angle [7,8,9,10]. However, research based on the graphene produced with the industrial standards shows that graphene is hydrophobic [11].

Nanoporous graphene is considered an ultimate inorganic material [12]. Studies have shown that porous single-layer graphene has excellent properties that result in high water flux and salt rejection capabilities [13,14,15]. Even though the application of single-layer graphene in the desalination process shows a sensational outcome, the preparation of the single-layer graphene membranes with a large surface area for its application process is problematic due to the formation of cracks and overlap of graphene sheets [16,17,18]. The use of multilayered graphene can be a good alternative to the above-mentioned issues faced by single-layer graphene [19,20,21]. However, the performance of the multilayered graphene is constrained by the number of graphene layers and its length [22]. Based on theory, the nanoporous graphene can reject ions and can deliver higher water flux in orders of magnitude 2 to 3 times than that of the membranes which are commercially available for reverse osmosis (RO) [23].

For a nanoporous graphene system, the pore size also plays an important role. Iwasaki et. al. [24] showed that at a pore diameter of 5.5 Å, the salt ions are rejected effectively while allowing the permeation of water molecules through the nanoporous single-layer graphene. In a recent study, the critical diameter of the nanopore in the multilayered graphene membrane is estimated to be 1.36 nm below which the dynamics of water confined inside the pore becomes abnormal and cannot be described by the Stokes–Einstein relationship [25]. The study of the implementation of functionalized graphene in the desalination process shows that effective salt rejection and water permeance can be achieved [14,22,26,27,28]. The selective passage of ions through the nanopore of the graphene membrane can be controlled by varying the functionalization of the nanopore, its size, and its electrostatic interaction with the walls of the membrane [22,23,27]. The improvement of the ability of the graphene to reject NaCl when it is functionalized is shown in a study using molecular dynamics (MD) simulations [13]. Graphene-based nanopores are used to mimic biological ion channel structures for ion-selective conduction using tunable voltage [29]. The increase in the strength of the applied electric field showed a significant increase in ion separation in a bilayer porous graphene membrane [30]. A recent study conducted to analyze the effect of various chemical functional groups on the performances such as salt rejection and water flux through functionalized nanoporous graphene showed the importance of employing a suitable arrangement of Alkyl functional groups for the membrane to be more efficient [31].

In this study, we have analyzed how the orientation of the layer in the multilayered porous graphene system affects the transport of water for two different pore shapes using molecular dynamics. The results of this study give a better insight into the effects of the orientation of the layers and the shape of the pore in multilayered porous graphene.

## 2. Model and Methods

The model of the multilayered nanoporous graphene system used in this study is shown in Figure 1. The multilayer was formed by tightly stacking five graphene sheets with a layer spacing of 3.5 Angstroms. Two reservoirs with TIP3P-EW [32] water molecules were used, one on each side of the graphene sheets. The size of the simulation system was 30 Å × 30 Å × 140 Å. The configuration of the simulation domain is given in Figure 1. Two different-shaped pores were used in this study. The first pore type is closer to a circular geometry, hence it is considered a circular pore, and the other pore that closely resembles a triangle is termed a non-circular (NC) pore in this study. The circular pore geometry had an effective diameter of 10 Å. The effective diameter refers to the circle with the maximum diameter that can be inscribed inside the pore. The non-circular pore used in this study had the pore size in which an effective equilateral triangle with a length of 10.40 Å can be inscribed inside the pore. The graphene layers with pores were modeled using SAMSON (Software for Adaptive Modeling and Simulation of Nanosystems, SAMSON version 2022 R1) software [33] and the structure was energy minimized using the FIRE (Fast Inertial Relaxation Engine) algorithm [34]. The molecular dynamics simulation was carried out using LAMMPS software (Lammps version 3 March 2020) [35]. The visualization of the simulations was carried out using used VMD (visual molecular dynamics) software (VMD version 1.9.4a55) [36]. A pressure difference of 150 MPa was applied to simulate the water transport. The pressure difference was achieved by applying ambient pressure to the graphene layer (which acts as a piston) present at the end of the permeate region and the desired pressure was applied to the graphene piston at the end of the feed region [31]. The simulation consisted of 2138 water molecules in total, of which 1710 water molecules were present in the feed region and 428 water molecules were present in the permeate region at the beginning of the equilibration run. Lennard–Jones potential was used to represent the interactions between atoms and the SHAKE algorithm [37] was used to keep the water molecules constrained. The canonical ensemble NVT and Nosé-Hoover thermostat [38] were used in this study. PPPM style (particle–particle particle–mesh) was used to treat the long-range electrostatic interactions between the atoms. The setup was equilibrated for 1 nanosecond and the production run was carried out for 7 nanoseconds with a timestep of 1 femtosecond. The cut-off distance for the truncation of the potential interactions used in this study was 10 Å. We used AIREBO (adaptive intermolecular reactive bond order) potential [39] for the interactions between the carbon atoms for the multilayered membrane. The interactions between the water molecules and the graphene were calculated using the Lorentz–Berthelot mixing rule.

In this study, we have determined how the angle orientation of different layers of graphene in the nanoporous multilayered graphene system influences the transport of the water molecules. To discard the influence of the entrance and exit effect of the membrane system, the first and last layers of the multilayered porous membrane system were kept unaltered. The different patterns used in the multilayered nanoporous graphene membrane are given in Table 1 and their representations are given in Figure 2c–f. We used 15, 30, and 45-degree tilt variations for the angled graphene sheets. The base case is the one with no tilt to any of the layers. In pattern 1, only the middle layer was tilted. In pattern 2, the 2nd and 4th layers were tilted, while in pattern 3, all layers other than the first and last layers were tilted. As mentioned earlier, we used two different pore shapes (circular and non-circular). As the transport of the water molecules varies based on the pore shapes [40], this is not a direct comparison between two pore shapes; rather, this study is aimed to get a better understanding of the variations of patterns and angles in both the circular and non-circular pore shapes.

## 3. Results and Discussion

Cumulative molecule passage (CMP) is the increase in the number of water molecules by successive addition of those water molecules that have completely passed through the membrane pore. Figure 3 shows the CMP of water molecules for various patterns and angles compared with the base case. At the end of the 7 ns simulation time, the membrane with a circular pore without any variation of the angle of the pore resulted in the transport of 792 water molecules. Pattern 1 membrane with a 45° showed a net increase of 16% higher water molecules with the total number of water molecules transported being 919 water molecules. This pattern is the highest performing pattern. A similar kind increase in the number of molecules being transported was also noticed in pattern 1 of the non-circular pore. The base case of the non-circular pore resulted in the transport of 404 water molecules. The non-circular pattern 1 pore membrane with 15° transported around 42% more water molecules with the total number of water molecules transported through the pore being 575. Interpreting the graphs in Figure 3, we can see that there is a significant impact on the number of water molecules transported by both the angle at which the graphene layer is oriented and also the pattern. In the circular pore case, pattern 1 with 15° and 45° degrees showed an increase in the number of water molecules transported. For the non-circular pore considered in this study, pattern 1 with 15° showed better water molecules transport. Pattern 1 has performed well in general, in which only the middle layer is oriented to some angle. Pattern 1 closely resembles an hourglass shape. In earlier studies, researchers have proved that the hourglass-shaped nanopore CNT (carbon nanotube) has better water molecule transport capabilities when compared to a normal CNT [41,42]. The key aspects for most of the configurations other than pattern 1 are less effective performance due to the reduction of the pore volume and the formation of a large energy barrier inside the pore due to its pore morphology. In patterns 2 and 3, for the circular pore configuration and for most of the non-circular pore configuration, we can see the significant presence of an empty state (no water molecule inside the pore). This can be easily visualized in Figure 3b–f showing that the increase of CMP stops and becomes constant. We can also see that there is a large difference between the CMP of circular and non-circular shaped pores, largely due to the pore morphology. The non-circular pore has a relatively lower accessible pore area than that of the circular pore configurations.

Figure 4 shows the free energy of occupancy fluctuations of the molecules inside the nanopore for different patterns and angles. Detailed discussions regarding the free energy of occupancy fluctuations have been carried out in many previous studies [43,44,45]. The free energy of occupancy fluctuations can be calculated using the formula –ln[P(N)], where P(N) is the probability of finding the exact number of water molecules inside the nanopore at a given time. The water occupancy inside the nanopore is determined by the local excess chemical potential, which is the negative free energy of removing a water molecule from the nanopore [45]. From Figure 4, we can notice that the most favorable number of water molecules for the base case of the circular pore is 18 with 115 occurrences and the most favorable number of water molecule occurrences for pattern 1 with 45° is 19 with 134 occurrences. Also, we can see that pattern 1 with 45° never had an empty state (an empty pore without any water molecules). The pore is either partially filled or in a filled state throughout the 7 ns simulation. The lowest number of molecules found inside this case is 8. This shows that the energy barrier that occurs in this pore configuration is relatively low compared to other cases. In the non-circular pore, the frequency of fluctuations and the amplitude of fluctuations of the number of water molecules in all cases were relatively large, which shows that the energy barrier inside the nanopore for these cases are relatively large. One of the key contributors to this large energy barrier is the assessable pore volume.

The radial distribution function of the oxygen atoms of water molecules with the first shell of the carbon atoms in the pore and the mass density of the water molecules inside the pore are given in Figure 5. Using the same force cutoff distance as in the simulation, the RDF is computed. Two atomic layers can be distinguished inside the nanopore from the two peaks in the radial distribution function plot [46]. The first and second distinctive peaks occur around 4.25 Å and 7.75 Å, respectively, for circular pores. For the non-circular pore, the peaks are not as distinctive as that of the circular pore; the first and second peaks are located at 4.25 Å and 8.25 Å, respectively. From the radial distribution function figure, we can observe that pattern 1 with 45° for the circular pore configuration and pattern 1 with 15° for the non-circular pore configuration show larger interaction between the water molecules and the carbon atoms of the pore, resulting in higher CMP. This also shows that more water molecules are accommodated inside the pore, which is further supported by the relevant mass density plot given adjacent to the relevant RDF plots.

The interaction energy between the carbon atoms and the water molecules determines the collective structure of the water molecules inside the pore [47]. They play a key role in the transport of water molecules through the membrane. Figure 6a,b shows the plot of the interaction energy between the carbon atoms that come in contact with the water molecules inside the pore along with the interaction force that is being exerted on the Z-direction for pattern 1 circular and non-circular cases. We can observe that the interaction energy in both the circular and non-circular base cases are relatively low and they show relatively low CMP. In addition, the cases circular pore with 45° orientation and non-circular pore with 15° orientation for pattern 1 show high interaction energy and high CMP. Similar kinds of observations were also made in water transport through nanotubes, in which the (6,6) nanotubes and (7,7) nanotubes had periodic arrangements starting with the hydrophobic and ending with the hydrophilic part. The (6,6) nanotubes with larger interaction energy performed better than the (7,7) nanotubes with relatively less interaction energy [42]. This shows that the interaction energy along with the resulting interaction force plays a key role in the transport of the water molecules through the pore. Furthermore, for the non-circular case with 30° and 45°, the 45° case shows lower interaction force than the case with 30° but the CMP of the 45° case is higher than the 30° case. This could be largely due to the influence of the interaction force that acts along the Z-direction. From Figure 6a,b, we can observe that the interaction force along Z-direction is larger for the 45° case when compared to the 30° case.

## 4. Conclusions

In this study, we have shown the importance of the orientation angle of the graphene layers and the pattern used in the construction of the multilayered nanoporous graphene for efficient water transport. Among the patterns used in this study, pattern 1 performed better when compared to the other patterns. In the circular pore configuration, pattern 1 with 15° and 45° and pattern 3 with 30° performed better than the base configuration (the one without any layer orientation). In terms of the non-circular pore, pattern 1 with 15°, 30°, and 45° performed better than the base configuration. The results of performance in terms of CMP for our study are well supported by the RDF and Interaction energy plots. Our study suggests that water transport through porous multilayered graphene can be augmented by altering the orientation of layers of the multilayered graphene membrane. Circular pore pattern 1 membrane with a 45° showed an augmentation of 16% in the number of water molecules transported, while the non-circular pattern 1 pore membrane with 15° showed an augmentation of 42% in the number of water molecules transported. Our findings may lead to better designs of multilayered nanoporous membranes with ultrafast water transport. The further study of pattern 1 with functionalized graphene multilayers could possibly lead to selective ion sieve membranes with high CMP.

## Figures and Tables

**Figure 1 micromachines-13-01786-f001:**
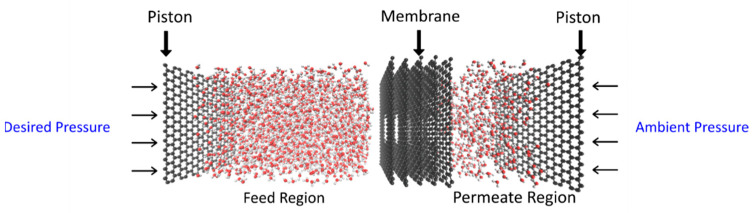
The computational domain.

**Figure 2 micromachines-13-01786-f002:**
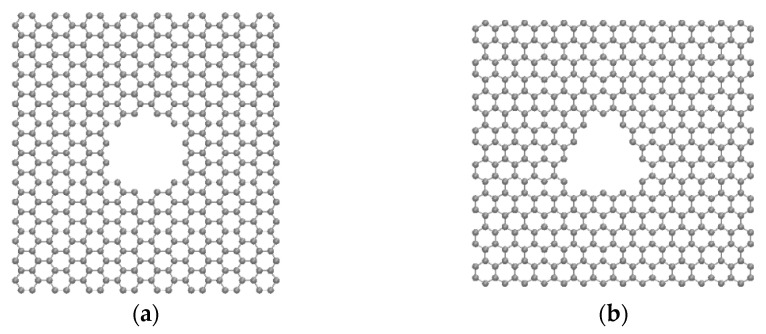
Pore shapes and patterns used in the simulation. (**a**) Circular shaped pore; (**b**) non-circular shaped pore; (**c**) representation of base pattern; (**d**) representation of pattern 1; (**e**) representation of pattern 2; (**f**) representation of pattern 3. Angled graphene sheets are represented in cyan color.

**Figure 3 micromachines-13-01786-f003:**
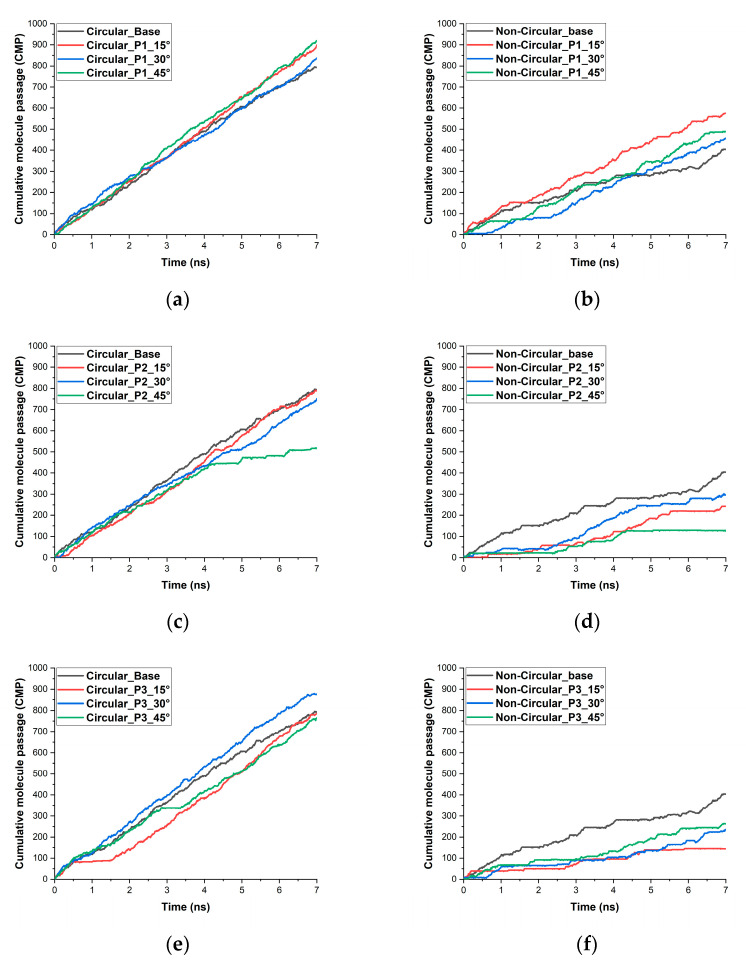
The cumulative molecule (water) passage through multilayered graphene nanopore (**a**) circular pore with pattern 1 (**b**) Non-circular pore with pattern 1 (**c**) circular pore with pattern 2 (**d**) Non-circular pore with pattern 2 (**e**) circular pore with pattern 3 (**f**) Non-circular pore with pattern 3.

**Figure 4 micromachines-13-01786-f004:**
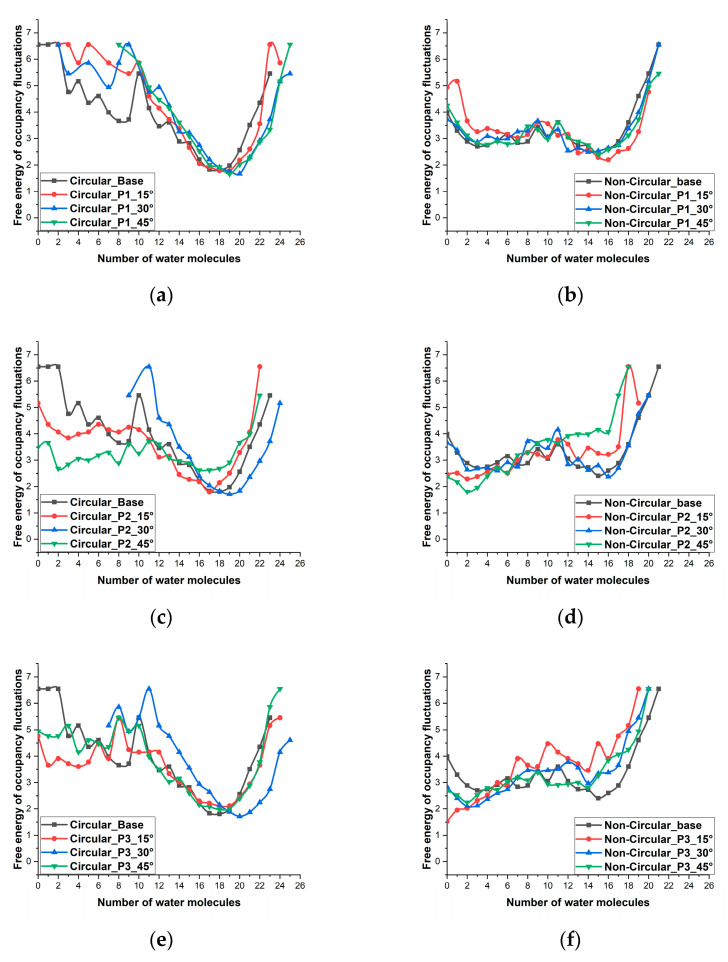
Free energy of occupancy fluctuations of water molecules inside the nanopore (**a**) circular pore with pattern 1 (**b**) Non-circular pore with pattern 1 (**c**) circular pore with pattern 2 (**d**) Non-circular pore with pattern 2 (**e**) circular pore with pattern 3 (**f**) Non-circular pore with pattern 3.

**Figure 5 micromachines-13-01786-f005:**
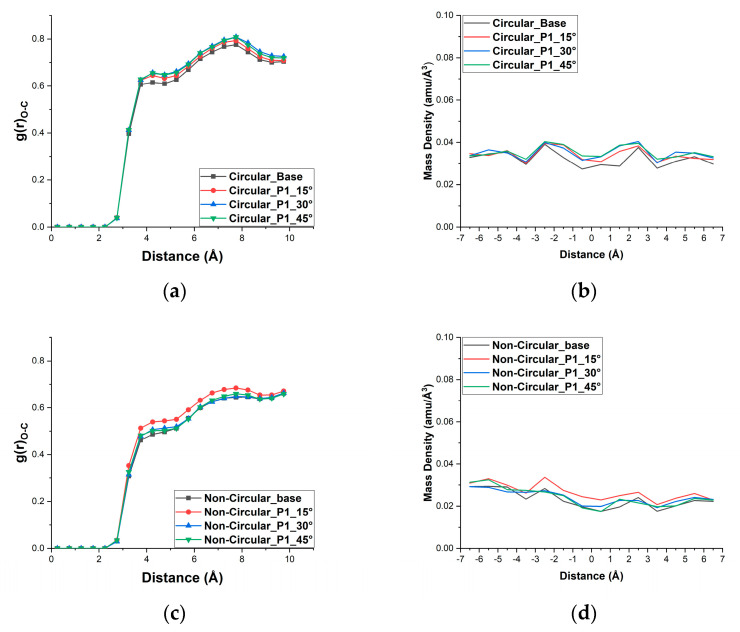
(**a**) Radial distribution function (RDF) of water molecules inside the circular pore with pattern 1 (**b**) density of water molecules inside the circular pore for pattern 1 (**c**) Radial distribution function (RDF) of water molecules inside the Non−circular pore with pattern 1 (**d**) density of water molecules inside the Non-circular pore for pattern 1.

**Figure 6 micromachines-13-01786-f006:**
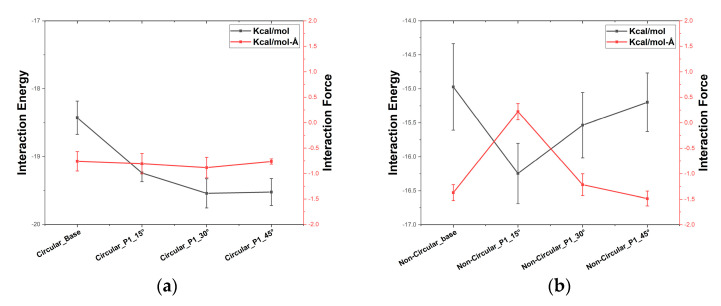
Interaction energy between the carbon atoms of the pore with the oxygen atoms of water molecules and the interaction force along Z−direction for the pattern 1 (**a**) circular porous membrane (**b**) non-circular porous membrane.

**Table 1 micromachines-13-01786-t001:** Shows the different patterns of graphene multilayer used in this study (θ—represents the angled graphene sheet).

Pattern	Layer 1	Layer 2	Layer 3	Layer 4	Layer 5
base	-	-	-	-	-
1	-	-	θ	-	-
2	-	θ	-	θ	-
3	-	θ	θ	θ	-

## Data Availability

The data presented in this study are available on request from the corresponding author. The data are not publicly available due to privacy and ethical restrictions.

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
