# Peer review of "Effect of Layer Orientation and Pore Morphology on Water Transport in Multilayered Porous Graphene"

_micromachines, 2022, doi:10.3390/mi13101786_

Round 1

Reviewer 1 Report

Dear Authors,

The manuscript is designed and structures good enough. There are few minor changes needs to amended before its consideration for the publication.

1.      The Introduction section is not adequate enough and references should have to be improvised. The entire first para only cites just one reference, but it introduces many different information to the reader without giving the proper references sources.

           /// Some recent studies claim that graphene is hydrophilic and exhibits a low water contact angle///// à but no reference is given.

2.      Likewise, the problem with the conventional martials are not properly discussed also there is no reference related to that.  This section has to be improvised.  

3.      The pore size of the circular as well as the NC case is not mentioned.

4.      The section 3 should to renamed as Results and Discussion as in general.

5.      Cumulative molecule passage should be a subsection 3.1 etc.

6.      The quality of all the figures should be improvised, the legends are not properly readable.

7.      All the figures SHOULD BE individually numbered and cited in the main text when necessary. For e.g., The figure 3. has six figures and all should be numbered from Figure 3a to Figure 3f and addressed individually, which is very much missing in the full manuscript.

8.      The response of CMP in connection with the time has reduced from Figure 3a to 3f. Fig.3a represents the circular case with P1 and Fig.3f represents the non-circular case with P3, the slope getting increased and why is that happens. Especially, in some cases (non-circular_P2_45O) the CMP response become constant, why. This kind of discussion should be added in the main text.

9.      The Figure 4 also be individually addressed and cited in the main text.

10.  //// From figure 4, we can notice that the most favorable number of water molecules for the base case of the circular pore is 18 with 115 instances of occurrences and the most favorable number of water molecule occurrences for pattern 1 with 45° is 19 with 134 occurrences //// à how the authors relate the number of molecules with the explicit number of instances. How the calculate it from the free energy fluctuation. Kindly explain it the revised version.

11.  ///We can observe that the interaction energy in both the circular and non-circular cases having relatively high water transport than in other cases is high.////à This lines are highly misleading please revise it. As I can get it from the figure 6a that interaction anergy is high for P1_15 than other cases but same is not true for NC case. But again this not properly explained in the manuscript.

12.  The authors failed to establish a relationship between the CMP, radial function and interaction energy and the plausible suggestion for the better stable structure among the discussed circular and NC cases. Please amend such an information in the manuscript.

13.  The conclusion also needs to be improvised.

Kind regards,

Reviewer.

Reviewer 2 Report

In this paper, water molecules transport in multilayered porous graphene was simulated by LAMMPS to explore the influence of pore morphology and layer orientation on transportation. Circular pore geometry and triangle (non-circular) morphology are considered as controllable variables. Pattern base and Pattern 1-3 are formed by the changes of the tilt angles of graphene sheets in different layers. Finally, the cumulative molecule passage and free energy of occupancy of water molecules are employed to evaluate each pattern and pore morphology effect on the transport of water molecules. The research showed that the non-circular pattern 1 pore membrane with 15° showed an augmentation of 42% in the number of water molecules transported.

Generally, the paper is well organized. However, it involved too much to be investigated. The paper may need substantial revision on the results and conclusion section for a deeper investigation. In addition, revisions may be needed for the following raised comments:

1)      Text for reference illustrations and pattern name may need to be consistent, for example fig.1, figure1, pattern 1 and pattern1.

2)      As shown in Fig.3, the cumulative number of water molecules transported through graphene with circular pore geometry is significantly higher than triangle morphology. Please explain this phenomenon in the discussion.

3)      The text description is for the four groups of different patterns of graphene multilayer set in Table.1. Could the author give the sketch diagram by VMD or OVITO?

4)      Please mark the figures in the same group by using (a) (b) (c)…

5)      The phenomenon in Fig.5 is not fully discussed. It is suggested to discuss the dependency of mass density of water molecules inside the pore on distance.

6)      There are typos/grammar error in the paper and the authors may need careful proofreading.

Round 2

Reviewer 2 Report

The authors addressed the raised comments and the paper is recommended for publication.